# Identifying the most effective essential medicines policies for quality use of medicines: A replicability study using three World Health Organisation data-sets

**Kathleen Anne Holloway**[1]*, **Verica Ivanovska**[2], **Solaiappan Manikandan**[3], **Mathaiyan Jayanthi**[3], **Anbarasan Mohan**[4], **Gilles Forte**[2], **David Henry**[5,6]

**1** Health and Nutrition, Institute of Development Studies, University of Sussex, Brighton, England, United Kingdom of Great Britain, **2** Essential Medicines and Health Products, World Health Organisation, Geneva, Switzerland, **3** Department of Pharmacology, Jawarhalal Nehru Institute of Medical Education and Research, Puducherry, India, **4** Department of Pharmacology, Government Theni Medical College, Theni, Tamil Nadu, India, **5** Institute for Evidence-based Healthcare, Bond University, Gold Coast, Queensland, Australia, **6** University of Melbourne, Melbourne, Australia

* kaholloway54@gmail.com

**Data Availability Statement:** All data used for the analyses shown in this article are shown in three supplementary excel files, labelled tables S1 Table

## Abstract

### Background

Poor quality use of medicines (QUM) has adverse outcomes. Governments' implementation of essential medicines (EM) policies is often suboptimal and there is limited information on which policies are most effective.

### Methods

We analysed data on policy implementation from World Health Organisation (WHO) surveys in 2007 and 2011, and QUM data from surveys during 2006–2012 in developing and transitional countries. We compared QUM scores in countries that did or did not implement specific policies and regressed QUM composite scores on the numbers of policies implemented. We compared the ranking of policies in this and two previous studies, one from the same WHO databases (2003–2007) the other from data obtained during country visits in South-East Asia (2010–2015). The rankings of a common set of 17 policies were correlated and we identified those that were consistently highly ranked.

### Findings

Fifty-three countries had data on both QUM and policy implementation. Forty policies were associated with effect sizes ranging from +13% to -5%. There was positive correlation between the composite QUM indicator and the number of policies reported implemented: (r) = 0.437 (95% CI 0.188 to 0.632). Comparison of policy rankings between the present and previous studies showed positive correlation with the WHO 2003–7 study: Spearman's rank correlation coefficient 0.498 (95% CI 0.022 to 0.789). Across the three studies, five policies were in the top five ranked positions 11 out of a possible 15 times: drugs available free at the

(QUM data), S2 Table (Policy data) and S3 Table (study comparison data) corresponding to the descriptions in the manuscript. The data used to compare policy impact on QUM in this study and the previous two published studies is also included (S3 Table).

**Funding:** No specific funding was sought or provided for this work

**Competing interests:** All authors have at some stage been involved with, or have worked for WHO. No other potential COI is declared. This does not alter our adherence to PLOS ONE policies on sharing data and materials.

point of care; a government QUM unit; undergraduate training of prescribers in standard treatment guidelines, antibiotics not available without prescription and generic substitution in the public sector.

## Interpretation

Certain EM policies are associated with better QUM and impact increases with co-implementation. Analysis across three datasets provides a policy short-list as a minimum investment by countries trying to improve QUM and reduce antimicrobial drug misuse.

## Introduction

Suboptimal (irrational, incorrect, inappropriate) use of medicines is widespread, wasteful, and causes poor patient outcomes including anti-microbial drug resistance [1–9]. Interventions to improve quality use of medicines (QUM) in low/middle-income countries have mostly been small-scale, of limited duration, with small to modest effects [10–11].

Evidence from studies that we conducted in public healthcare sectors in developing and transitional countries suggests that implementation of WHO essential medicines (EM) policies is associated with better quality use (rational use) of medicines (QUM), including more appropriate use of anti-microbial agents [12–14]. The original WHO global data-set [12] covered the period 2003–2007 and there was uncertainty about how well EM policies were executed (based on country self-reports), with simultaneous deployment of multiple policies making it difficult to estimate individual impacts. We accessed a second source of data collected during 2-week visits to countries in South-East Asia during 2010–15, where policy implementation was observed independently [14]. The analyses of these data confirmed several of the findings of the earlier studies [12–13], including a correlation between the total numbers of EM policies implemented and composite measures of QUM. However, it remains unclear which policies are associated with the largest beneficial effects on medicines use.

The aims of the present work were to analyse an updated global WHO data-set (2007–2011), which included some policies not previously evaluated, and to test the consistency of our earlier findings of an increased impact with larger numbers of implemented EM policies. In addition, we wished to assess replicability of findings by correlating the rankings of policies that were common to the three studies to determine whether certain policies were consistently associated with the largest effects.

## Methods

The analytical methods used have been described previously [12–14] and are summarised briefly here. QUM data (outcomes) were extracted from independent survey reports contained within the WHO medicines use database for the period 2006–2012 [3] and reported policy implementation data were obtained from WHO policy databases of surveys sent to Ministries of Health in 2007 and 2011 [15–16]. A dataset was created with one set of QUM and policy indicators for each country. Where the same QUM indicator was measured by more than one survey in the same country during 2006–2012, an average value was calculated. Where the same policy was reported differently in 2007 and 2011, the policy information reported from within one year of QUM survey was used or if this was not possible the data were excluded.

## Indicators

Thirteen QUM indicators were extracted from the WHO medicines use database [3]. Only surveys using recommended validated measures estimated from at least 600 prescriptions and/or three or more facilities were included [17–18]. The QUM indicators were all expressed as proportions and are described in Table 1, together with the directionality of better (or worse) QUM. Ten indicators were used in the previous analysis of WHO data [12].

Fifty-two indicators of reported policy implementation were extracted (49 from WHO questionnaires sent to Ministries of Health in 2007 and 2011 [19] and 3 from the WHO medicines use database [3]). The selected EM policies included all those that had been associated with better QUM in the previous two studies [12–14]; all were categorised as yes/no variables. Policies were excluded from analysis if there were fewer than six countries reporting implementation or non-implementation of the policy as was done previously [12].

Where policy indicators overlapped, only one was included. Where there was more than one indicator with a time-frame we included the one with the largest sample size. Through this process the number of policy indicators was reduced to 40 (reasons given in Table 2).

Eight policy indicators had not been analyzed in previous studies: availability of essential medicine list (EML) booklets at health facilities; existence of national legislation on drug promotion; prohibition of advertising prescription-only medicines to the public, a national task force to contain antimicrobial resistance and four policies concerning pharmacists—undergraduate training on the EML and standard treatment guidelines (STGs), continuing professional development and whether pharmacists prescribed in primary care (Table 2).

## Analyses

As previously [12–14], we did not try to perform head-to-head comparisons of different policies. Countries implemented different combinations of policies, so the impact of a single policy could not be separated from those that were co-implemented.

**Table 1. Indicators of Quality use of Medicines (QUM) and direction of better use.**

|    | Variable Name | Direction of better use |
|----|---------------|-------------------------|
| 1  | % patients prescribed antibiotics | Less |
| 2  | % patients not needing antibiotics that are prescribed them | Less |
| 3  | % upper respiratory tract infection cases treated with antibiotics | Less |
| 4  | % pneumonia cases treated with an appropriate antibiotic | More |
| 5  | % diarrhoea cases treated with antibiotics | Less |
| 6  | % diarrhoea cases treated with oral rehydration solution | More |
| 7  | % diarrhoea cases treated with anti-diarrhoeal drugs | Less |
| 8  | % malaria cases treated with an appropriate anti-malarial** | More |
| 9  | % prescribed drugs belonging to the Essential Medicines List | More |
| 10 | % drugs prescribed by generic name | More |
| 11 | % patients prescribed vitamins (mainly B complex & multivitamin) | Less |
| 12 | % patients prescribed injections | Less |
| 13 | % patients treated in compliance with standard treatment guidelines | More |

* Thirteen standard medicines use indicators [17–18] expressed as proportions and reported in surveys in more than 8 countries during 2006–2012.

** One indicator (% patients treated with an appropriate anti-malarial) was not used in any of the previous studies [12–14]. However, assuming that overall measurement of QUM will be more robust with more individual QUM indicators, and due to the large number of studies measuring antimalarial use in recent years, it was decided to include this extra QUM indicator on antimalarial use in this study.

**Table 2. Medicines policies hypothesised to improve quality use of medicines (QUM).**

| | Educational policies | Inclusion/exclusion from analysis with reasons | Whether policy was measured in one or both of previous two studies* |
|---|---|---|---|
| 1 | Public education on medicines use in the last two years | Included | Yes |
| 2 | Undergraduate training of doctors on the national Standard Treatment Guidelines (STGs) | Included | Yes |
| 3 | Undergraduate training of pharmacists on the national STGs | Included | No |
| 4 | Undergraduate training of doctors on the national Essential Medicines List (EML) | Included | Yes |
| 5 | Undergraduate training of pharmacists on the national EML | Included | No |
| 6 | Mandated continuing medical education that includes quality use of medicines (QUM) for doctors | Included | Yes |
| 7 | Mandated continuing medical education that includes QUM for pharmacists | Included | No |
| 8 | Mandated continuing medical education that includes QUM for nurses and/or paramedical staff | Included | Yes |
| | **Managerial policies** | | |
| 9 | Availability of Essential Medicines List booklet at health public** (from patient care indicators) | Included | No |
| 10 | Availability of Standard Treatment Guidelines booklet at health public** (from patient care indicators) | Included | Yes |
| 11 | Better drug supply** (as indicated by better drug availability from patient care indicators) | Included | Yes |
| 12 | National Essential Medicines List (EML) updated in the last five years | Excluded, as insufficient numbers of country responded "no" to make a comparison | |
| 13 | National Essential Medicines List (EML) updated in the last two years | Included | Yes |
| 14 | National Formulary updated in the last five years | Included | Yes |
| 15 | National Formulary updated in the last two years | Excluded, as duplicative of the policy on formulary updated in last 5 years | |
| 16 | National Standard Treatment Guidelines (STGs) updated in the last five years | Excluded as duplicative of the policy on national STGs updated in the last 2 years and more even distribution of countries with & without the policy | |
| 17 | National Standard Treatment Guidelines updated in the last two years | Included | Yes |
| 18 | Prescription audit done any time in the past | Excluded, as prescription audit in the last two years was felt to be more indicative of active policy | |
| 19 | Prescription audit in the last two years | Included | Yes |
| 20 | Generic prescribing policy in public sector | Included | Yes |
| 21 | Generic substitution in public sector | Included | Yes |
| | **Regulatory policies** | | |
| 22 | Active monitoring of Adverse Drug Reactions (ADRs) | Included | Yes |
| 23 | Antibiotics generally NOT available over-the-counter (OTC) (never/occasional = No; always/frequently = Yes) | Included | Yes |
| 24 | Injections generally NOT available over-the-counter (never/occasional = No; always/frequently = Yes) | Included | Yes |
| 25 | National legislation on drug promotion | Included | No |
| 26 | Co-regulation of drug promotion by government and industry | Included | Yes |
| 27 | Pre-approval of adverts for over-the-counter (OTC) medicines undertaken | Included | Yes |
| 28 | Existence of guidelines for the advertising of OTC medicines | Excluded as very few countries had such guidelines and this policy is partially duplicative of the policy on pre-approval of OTC drug adverts | |

*(Continued)*

**Table 2.** (Continued)

| | Educational policies | Inclusion/exclusion from analysis with reasons | Whether policy was measured in one or both of previous two studies* |
|---|---|---|---|
| 29 | Prohibition of advertising of prescription-only medicines to the public | Included | No |
| | **Structural policies** | | |
| 30 | Existence of a National Medicines Policy document | Excluded, as insufficient numbers of country responded "no" to make a comparison | |
| 31 | National medicines policy implementation plan | Included | Yes |
| 32 | National Ministry of Health (MOH) unit on promoting rational use of medicines | Included | Yes |
| 33 | Presence of National Drug Information Centre | Included | Yes |
| 34 | National strategy to contain antimicrobial resistance (AMR) | Included | Yes |
| 35 | National task force to contain AMR | Included | No |
| 36 | National reference laboratory for AMR | Excluded, as duplicative of other policies on antimicrobial resistance containment | |
| 37 | Drug and Therapeutic Committee (DTC) in half or more of all referral hospitals | Included | Yes |
| 38 | Drug and Therapeutic Committee in half or more of all general hospitals | Included | Yes |
| 39 | Drug and Therapeutic Committee in half or more of all provinces | Excluded, as duplicative of DTCs in general hospitals | |
| 40 | Ministry of Health regulation to have Drug and Therapeutic Committees | Excluded, as duplicative of other DTC policies | |
| | **Economic policies** | | |
| 41 | All drugs on the national Essential Medicines List (EML) provided free of charge in a national health or social insurance system | Included | Yes |
| 42 | Drugs dispensed free of charge to pregnant women | Excluded as partially duplicative of drugs dispensed free of charge to children and not measured in previous studies | |
| 43 | Drugs dispensed free of charge to the poor | Included | Yes |
| 44 | Drugs dispensed free of charge to children under five years | Included | Yes |
| 45 | Drugs dispensed free of charge to the elderly | Excluded as duplicative of other free drug policies | |
| 46 | NO Drug sales revenue used to supplement prescriber income | Included | Yes |
| 47 | NO user fees for medicines | Included | Yes |
| 48 | NO fees for consultation or registration | Included | Yes |
| 49 | Prescribers dispense in the public sector | Excluded as the number of countries with this policy was small and the policy indicator does not address the important issue of prescribers who earn money from drug sales generally in the private sector. In addition, it was not measured in previous studies. | |
| | **Human resource management policies** | | |
| 50 | Prescribing by pharmacists in public primary care | Included | No |
| 51 | No prescribing by staff with less than one month's training in public primary care | Included | Yes |
| 52 | Prescribing by nurses and/or paramedical staff in public primary care | Included | Yes |

* Includes all policies found to be associated with improved QUM as found in previous studies [12–14].

** Patient care indicators extracted from the Medicines Use Database and where the countries with values above the median across countries are classified as having better implementation of national STGs/EML and drug supply respectively.

For each QUM indicator we calculated the mean difference (expressed as a percentage) between countries reporting implementation (or not) of specific policies. For each policy, we estimated the average difference across all 13 QUM indicators, aligning directionality of better use (positive number) and worse use (minus number), and including only those QUM indicators where there were at least three countries with and three without the policy in question [12].

To assess the impact of multiple policies we generated a composite QUM score from 13 QUM indicators, which enabled all countries to be included in the analysis [12]. We calculated how far each country's value lay above or below the mean value from all countries for each QUM indicator expressed as standard deviation (SD) units. We then calculated the average of the SD unit increments across the thirteen QUM indicators for each country and used linear regression to assess correlation with the number of EM policies that were implemented [12]. We limited these analyses to policies that had a statistically significant association with better QUM in the univariate analyses.

Individual QUM indicators were also regressed on the number of implemented policies to determine whether specific aspects of QUM were influenced by the intensity of policy implementation. The impact of country wealth was assessed by including Gross National Income per capita [20] in multiple linear regression analyses and by repeating the regression analyses for countries with GNIpc above and below the median of USD 2315.

## Testing the replicability of findings across three studies

Statistical analysis methods used in the present study were the same as those used in the earlier WHO analysis [12–13] and the SE Asia country visit analysis [14], enabling us to compare findings across three studies [12–14]. For each of the three data-sets we ranked the policies based on their estimated impact from the univariate analyses. We used non-parametric regression analysis to measure the correlation between the ranking of the policies that were common to the three studies [12–14]. We established the overall ordering of policies by calculating the sum of their ranks across the 3 studies. All analyses were done in Excel 2016, using either Epi Info (version 7.2.1.0)- or Stats Direct (version 3.1.20).

All work involved secondary analyses of data collected for other purposes. Data were aggregated at the level of countries or policies, not individuals, so research ethics board approval was not required.

## Results

Fifty-three countries had data on both QUM and policy implementation. Regional distribution of countries was Africa (23), Eastern Mediterranean (7), Europe (2), Latin America (2), South-East Asia (11) and Western Pacific (8). On average, data were available from a median of 2 (range 1–30) QUM surveys and 4 QUM indicators (range 1–13) per country. Each QUM indicator was used by a median of 19 countries (range 9–37). Out of a potential 2120 policy responses (40 policies in each of 53 countries), 1787 (84%) were available for analysis. Of fourteen countries reporting policies in both 2007 and 2011, 85 (18%) responses out of a potential 476 policy responses (34 policies [measured in both 2007 and 2011] x 14 countries) were reported differently and of these 54 (11%) were excluded. Supporting information (S1 Table) describes the 13 QUM indicators and 3 policy indicators obtained from the WHO medicines use database, by country together with the survey references. Supporting information (S2 Table) describes information on the reported implementation of 52 policies by country. Supporting information (S3 Table) describes the impact of common policies in this study and the two previously published studies–this being the data used in the replicability analysis.

## Strength of associations for individual policies

Table 3 shows the estimates of policy effect on QUM by policy type, comparing results in countries that did, or did not, report implementation. Fig 1 shows these results in order of their estimated effect size.

Policies that were statistically significantly associated with 5–10% or higher effects on QUM included: drugs free at the point of care for children less than five years and the poor; generic substitution; a national strategy to contain antimicrobial resistance; a national body dedicated to QUM; booklets of the national essential medicines lists and standard treatment guidelines available at health facilities; not having systemic antibiotics available over-the-counter; an updated national formulary; no prescriber revenue from medicine sales; national legislation on drug promotion; public education; all drugs on the national Essential Medicines List (EML) provided free of charge in a national health or social insurance system; drug and therapeutic committees in hospitals; and undergraduate education of doctors and pharmacists on standard treatment guidelines.

Of the 27 policies that were associated with positive effects, the average estimated effects were: 9.3% (range 7.0 to 13.0)% for economic policies; 7.4% (2.3 to 10.5%) for managerial policies; 7.3% (range 5.6 to 10.2%) for structural policies; 4.9% (range 2.3 to 6.8%) for educational policies; 4.9% (range 1.7 to 8.6%) for regulatory policies, and 4.2% (range 3.2 to 5.1%) for human resource management policies.

## Impacts of multiple policies and national wealth

Fig 2 shows a scatter-gram of composite QUM scores versus the number of policies reported implemented. Correlation between the composite QUM indicator and the number of significantly effective policies reported as implemented (out of 18) was moderate (r = 0.437; 95% CI 0.188 to 0.6322) and strengthened when regression was limited to countries with more than two QUM indicators (r = 0.510; 95% CI 0.243 to 0.704). Inclusion of a national wealth measure (GNIpc) in the regression had no effect (r = 0.51; 95% CI 0.243 to 0.704) and the correlation coefficients were similar when analyses were conducted separately for countries with GNIpc levels above (r = 0.55, p = 0.018) and below the group median (r = 0.41, p = 0.048)).

When we examined the impact of multiple policies on individual QUM indicators (Supporting information S4 Table) we found that the percentage of all cases treated with antibiotics was significantly less with implementation of a greater number of policies (r = -0.375; 95% CI -0.624 to -0.059) as was the percentage of upper respiratory tract infection cases treated with antibiotics (r) = -0.554; 95% CI -0.796 to -0.161. Fig 3 shows a scatter-gram of the percentages of upper respiratory tract infection cases treated with antibiotics versus the number of policies reported implemented. The differences in the percentage of upper respiratory tract infection cases treated with antibiotics were large, ranging from 80–100% in countries implementing less than four EM policies to 30–70% in countries implementing more than 15 policies.

## Replicability of effects across studies

Table 4 summarises the results for 17 policies that were common to the three studies ordered by the sum of the ranks across the three studies. The table also provides the individual study rankings and whether the univariate analyses of effect sizes had 95% CI that excluded zero. We found a significant correlation between the ranking (24 common policies) in the present study (2007–2011) and that found in the previous analysis of WHO global data (2003–2007): rank correlation coefficient Rho = 0.498 (95% CI 0.022 to 0.789). Correlation between the ranking (20 common policies) in the current analysis and that from the SE Asia country visits was weaker: Rho = 0.465 (95% CI -0.020 to 0.773). Nine policies had effect sizes of 4–10% that were

**Table 3. Difference in medicines use across 13 QUM indicators between countries reporting implementation / non-implementation of 40 essential medicines policies.**

| Average difference across all QUM indicators where number of countries per QUM indicator per arm of policy implementation is >2 countries | No. QUM indicators in av. diff. calculation | Average (Av.) difference (diff.) in QUM with 95% CI | Whether policy included in variable on number of EM policies implemented* |
|---|---|---|---|
| **EDUCATIONAL POLICIES** | | | |
| 1 | Public education on medicines use in the last two years | 13 | 6.8 (4 to 10) | Yes |
| 2 | Undergraduate training of pharmacists on the national Standard Treatment Guidelines (STGs) | 12 | 6.3 (2 to 11) | Yes |
| 3 | Undergraduate training of doctors on the national STGs | 12 | 5.4 (2 to 9) | Yes |
| 4 | Undergraduate training of doctors on the national Essential Medicines List (EML) | 12 | 3.8 (-1 to 9) | No |
| 5 | Undergraduate training of pharmacists on the national EML | 12 | 2.3 (-3 to 7) | No |
| 6 | Continuing medical education of pharmacists | 13 | -0.8 (-7 to 5) | No |
| 7 | Continuing medical education of doctors | 13 | -2.4 (-8 to 3) | No |
| 8 | Continuing medical education of nurses and/or paramedical staff | 13 | -5.1 (-14 to 4) | No |
| **MANAGERIAL POLICIES** | | | |
| 9 | Generic substitution in public sector | 11 | 10.5 (3 to 18) | Yes |
| 10 | Availability of Essential Medicines List booklet at health public** (from patient care indicators) | 9 | 10.3 (4 to 16) | Yes |
| 11 | Availability of Standard Treatment Guidelines booklet at health public** (from patient care indicators) | 10 | 9.8 (1 to 19) | Yes |
| 12 | National Formulary updated in the last five years | 11 | 8.2 (3 to 14) | Yes |
| 13 | Prescription audit in the last two years | 5 | 5.5 (-5 to 16) | No |
| 14 | Better drug supply** (as indicated by better drug availability from patient care indicators) | 13 | 5.0 (-3 to 13) | No |
| 15 | Generic prescribing policy in public sector | 13 | 2.3 (-5 to 10) | No |
| 16 | National Essential Medicines List (EML) updated in the last two years | 11 | 0.9 (-3 to 5) | No |
| 17 | National Standard Treatment Guidelines (STGs) updated in the last two years | 13 | -3.3 (-8 to 2) | No |
| **REGULATORY POLICIES** | | | |
| 18 | Antibiotics generally NOT available over-the-counter (OTC) (never/occasional = No; always/frequently = Yes) | 5 | 8.6 (2 to 16) | Yes |
| 19 | National legislation on drug promotion | 12 | 6.8 (1 to 12) | Yes |
| 20 | Injections generally NOT available over-the-counter (OTC) (never/occasional = No; always/frequently = Yes) | 9 | 0.0 (-9 to 9) | No |
| 21 | Prohibition of advertising of prescription-only medicines to the public | 4 | 2.5 (-13 to 18) | No |
| 22 | Active monitoring of Adverse Drug Reactions (ADRs) | 13 | 1.7 (-4 to 8) | No |
| 23 | Co-regulation of drug promotion by government and industry | 7 | -0.5 (-7 to 6) | No |
| 24 | Pre-approval of adverts for over-the-counter (OTC) medicines undertaken | 7 | -2.4 (-9 to 5) | No |
| **STRUCTURAL POLICIES** | | | |
| 25 | National task force to contain AMR | 6 | 11.1 (0 to 23) | Yes |
| 26 | National strategy to contain antimicrobial resistance (AMR) | 11 | 10.2 (5 to 16) | Yes |
| 27 | National Ministry of Health (MOH) unit on promoting Quality Use of Medicines (QUM) | 10 | 9.8 (3 to 17) | Yes |
| 28 | Drug and Therapeutic Committee in half or more of all general hospitals | 11 | 7.3 (0 to 15) | Yes |
| 29 | Drug and Therapeutic Committee (DTC) in half or more of all referral hospitals | 13 | 5.6 (1 to 11) | Yes |

*(Continued)*

**Table 3.** (Continued)

| Average difference across all QUM indicators where number of countries per QUM indicator per arm of policy implementation is >2 countries | | No. QUM indicators in av. diff. calculation | Average (Av.) difference (diff.) in QUM with 95% CI | Whether policy included in variable on number of EM policies implemented* |
|---|---|---|---|---|
| | EDUCATIONAL POLICIES | | | |
| 30 | Presence of National Drug Information Centre | 12 | 0.6 (-8 to 9) | No |
| 31 | National medicines policy implementation plan | 12 | -3.5 (-15 to 8) | No |
| | ECONOMIC POLICIES | | | |
| 32 | Drugs dispensed free of charge to the poor | 12 | 13.0 (6 to 20) | Yes |
| 33 | Drugs dispensed free of charge to children under five years | 12 | 12.2 (5 to 19) | Yes |
| 34 | NO Drug sales revenue used to supplement prescriber income | 13 | 7.9 (2 to 14) | Yes |
| 35 | All drugs on the national Essential Medicines List (EML) provided free of charge in a national health or social insurance system | 12 | 6.3 (3 to 9) | Yes |
| 36 | NO user fees for medicines | 12 | 7.0 (-2 to 15) | No |
| 37 | NO fees for consultation or registration | 7 | 0.0 (-6 to 6) | No |
| | HUMAN RESOURCE MANAGEMENT POLICIES | | | |
| 38 | Prescribing by pharmacists in public primary care | 13 | 5.1 (-3 to 14) | No |
| 39 | No prescribing by staff with less than one month's training in public primary care | 11 | 3.2 (-4 to 11) | No |
| 40 | Prescribing by nurses and/or paramedical staff in public primary care | 8 | -5.1 (-11 to 1) | No |

* The variable on the number of policies reported implemented was adjusted for missing data as follows: adjusted policy number = (number of policies reported/(N- number of missing values for policies)) x N, where N was the number of effective policies [12].

statistically significant in two or more of the three studies. Five policies had consistently high positions in the orderings (highlighted in Table 4), appearing in the top 5 ranked positions 11 out of a possible 15 times. They were: medicines free at the point of care; the presence of a government QUM unit, undergraduate training of prescribers in STGs, antibiotics not available without prescription and generic substitution allowed in the public sector. Statistically significant better QUM associated with implementation of more policies was seen in all three studies [12–14].

## Discussion

The main findings from the current study of the most recent WHO data-bases were three-fold. Firstly, some essential medicines policies were associated with better QUM. The strongest associations were for: medicines free at the point of care, implementation of STGs and the EML, a national body to promote QUM, a national strategy to contain AMR, disallowing antibiotic availability OTC, generic substitution in the public sector, hospital DTCs, and public education. Secondly, all policy categories had similar overall degrees of association with better QUM. Thirdly, there was a positive correlation between the number of policies that countries reported implementing and their measures of QUM.

The WHO data have significant limitations, notably the reliance on self-report and the variable co-implementation of several policies, making it difficult to discern the true effects of individual policies. In addition, multiple policies and QUM measures make chance associations likely and limit the interpretation of statistical significance testing. In this situation a consistent finding of a relationship between intensity of policy implementation (number of policies) and a composite measure of QUM is important. In this and previous studies [12–14], there were

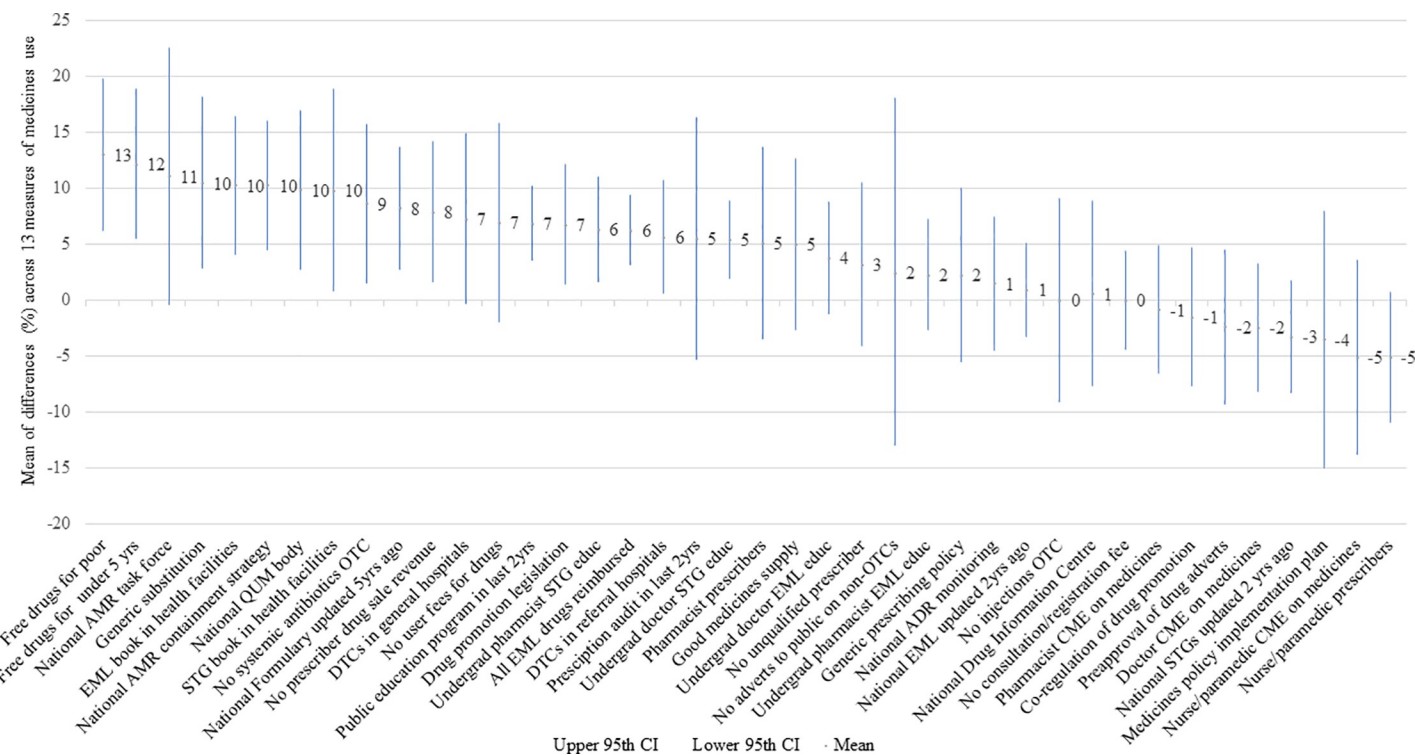

**Fig 1. Differences in quality use of medicines between countries that did versus did not report implementation of specific medicine policies.** Bars and numbers represent the estimated mean effect and 95% CI for the mean effect of each policy on a composite measure of QUM. X-axis acronyms: AMR = antimicrobial resistance; EML = Essential Medicines List; QUM = Quality Use of Medicines; STG = Standard Treatment Guideline; OTC = Over-the-Counter; DTC = Drug and Therapeutic Committee; ADR = Adverse Drug Reaction; CME = Continuing Medical Education.

moderate associations between implementation of more EM policies and better QUM, as reflected by both a composite QUM indicator and individual QUM indicators, notably lower antibiotic use in upper respiratory tract infection. The strength of association seen in this study was like those seen in the previous analyses of WHO global data [12] and data from SE Asia [14]. Unlike the previous two studies the association between EM policies and QUM appeared to be weaker in poorer countries than in wealthier ones, although the association was stronger when regression analysis was limited to more robust QUM data (based on more than 2 QUM indicators).

Although analyses of multiple policy exposure are valuable, these analyses have their own limitations. Most importantly, the exposure variable is the number of equally weighted policies and this does not assist in the identification of the most effective policies. With potentially large numbers of policies and co-variates, and modest number of countries, it was not possible to perform multi-variable analyses and conduct comparisons of individual policies. For these reasons, we assessed the replicability of the ordering of policies by estimated effects across the three studies we have completed. The correlations of the rank orders between the present and previous analyses were modest when measured across the full set of 17 policies that were common to each study. However, the five highest ranking policies (Table 4) occupied the top five places on 11 out of a possible 15 occasions.

In a situation defined by weak data we think the replicability we found across three separate studies, using almost identical methods, is the strongest evidence for identifying the most effective essential medicines policies. We are not suggesting that these policies are the only ones that should be considered for implementation. Countries with particular needs may

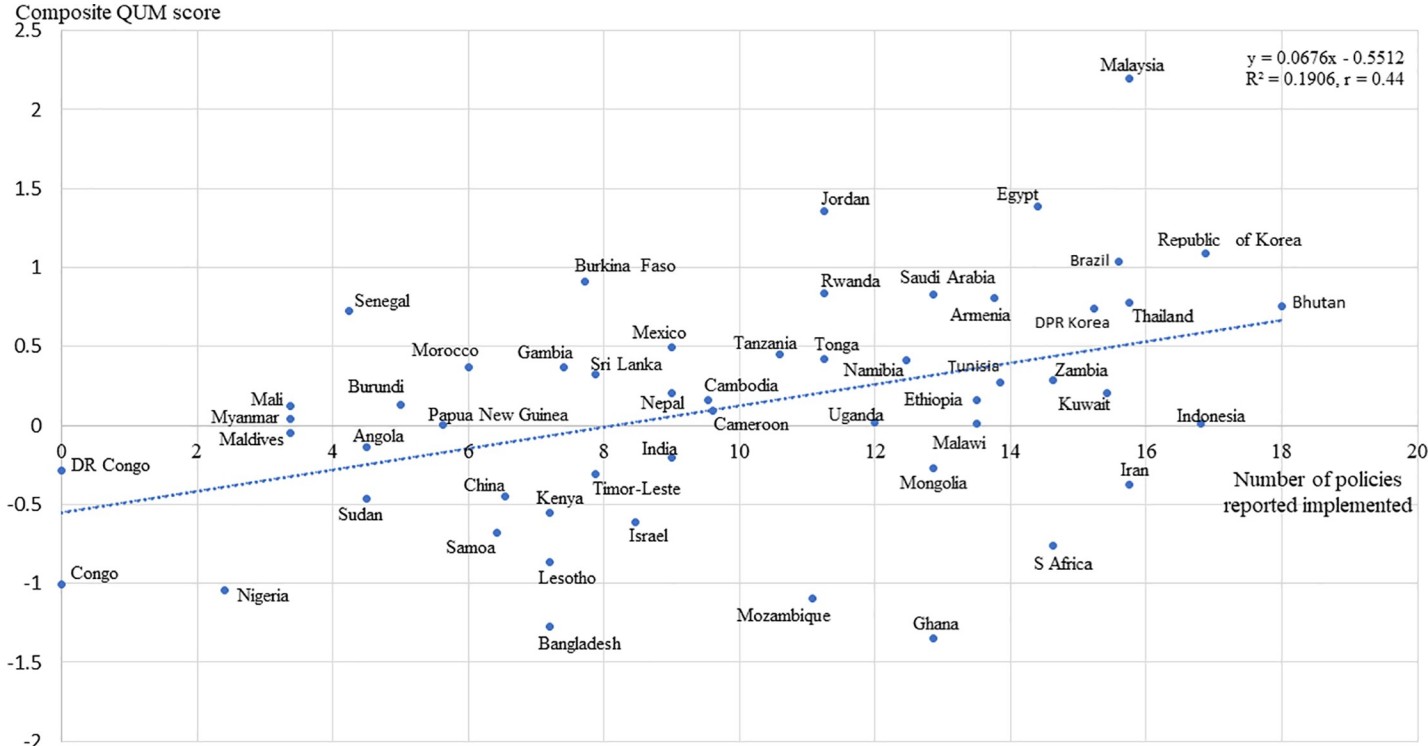

**Fig 2. Scatter-gram of the composite QUM indicator score versus the number of policies reported implemented.** Data is good enough to show better QUM with implementation of more policies, but not to benchmark country performance.

choose from a larger basket of policies. However, five apparently strong policies, each from a different category, represents a minimum investment for countries seeking to improve QUM and optimize the consumption of antimicrobial drugs. The policies are: drugs free at the point of care; existence of a government QUM unit; undergraduate prescriber training in standard treatment guidelines; antibiotics not available over the counter without prescription and generic substitution allowed in the public sector. Because they come from different policy categories it is possible that they have complementary effects, although that couldn't be tested here.

## Comparison with the broader literature

Previous reviews have recommended implementing similar policies to improve QUM [21–22]. Other studies reporting on actual policy effectiveness reported on: prescriber education [10–11]; public education [23–24]; an MOH body dedicated to promoting QUM [25]; hospital drug and therapeutic committees (DTCs) [26]; non-allowance of prescriber revenue from medicine sales [27–29]; non-allowance of antibiotic availability OTC [30], and national legislation and monitoring of drug promotional activities [31]. Greater effectiveness of multi-faceted interventions (which may involve multiple localised policies), as opposed to single-faceted ones, has also been found elsewhere [32–34]. Furthermore, the better QUM seen here with implementation of more policies was large and comparable with intervention effects reported elsewhere [10–11, 32–34]. However, the sustainability of the better QUM achieved with national medicines policy implementation is likely to be much greater than that achieved with the discrete interventions implemented locally.

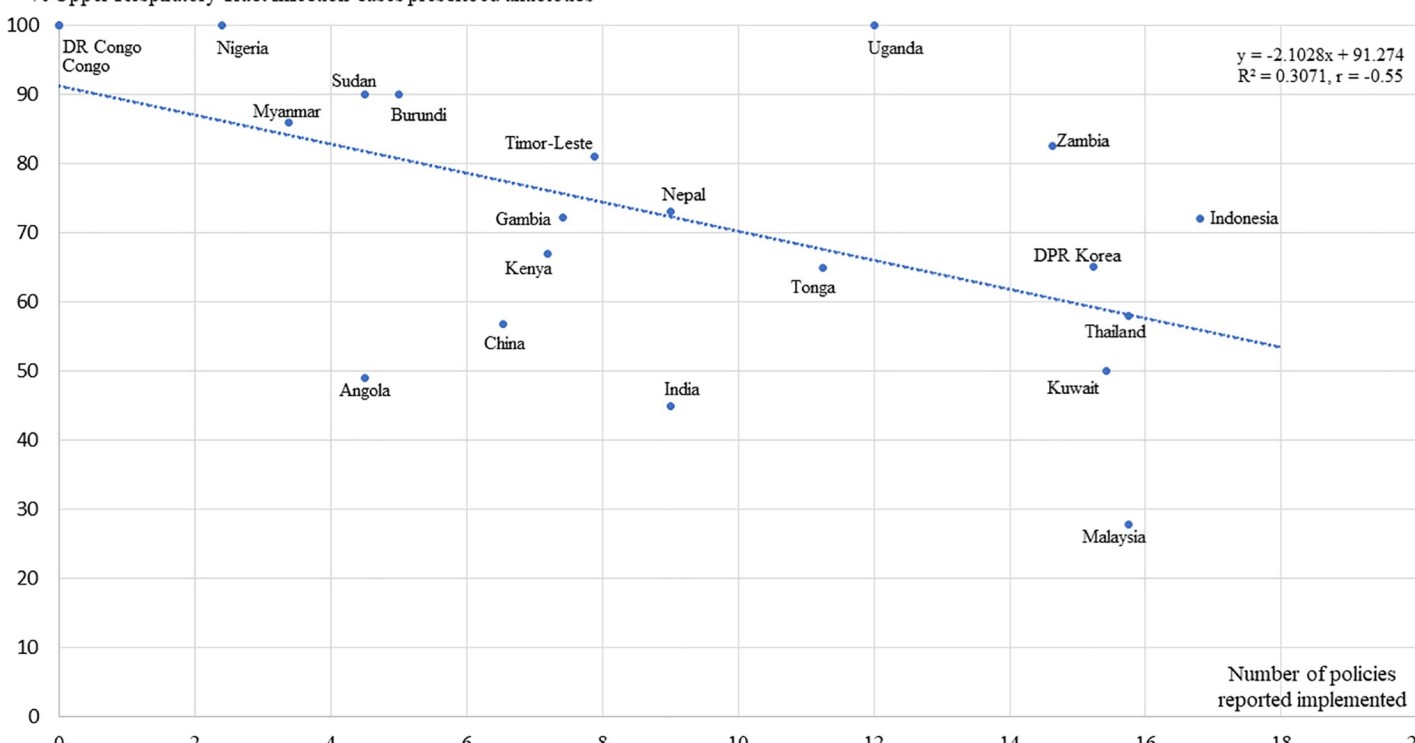

**Fig 3. Scatter-gram of the % upper respiratory tract infection cases treated with antibiotics versus the number of policies reported implemented.** Data is good enough to show less antibiotic use in upper respiratory tract infection with implementation of more policies, but not to benchmark country performance.

### Limitations

We have extensively discussed the limitations of the WHO data-bases above and in previous reports [12–13]. The policy data used are reliant on self-reports of implementation, which may have been inaccurate, and the apparent effectiveness of individual policies may have been due to co-interventions. The small number of countries, large numbers of possible policy combinations, and other factors, including political will and economic stability, can reduce implementation effectiveness and hamper attempts to estimate the impacts of individual policies or specific policy combinations. Another weakness of the data was the assumption that policies may have remained the same over time. However, the fact that there were only small differences between estimates from countries reporting policy implementation for both 2007 and 2011 suggests that most policies generally remained constant. Furthermore, misreporting and misclassification would likely have weakened any associations seen between policy implementation and QUM.

There were weaknesses in the QUM data. Firstly, they come from surveys published in the literature. While only surveys using standard methodology and indicators [17–18] were used, they were often based on small sample sizes. Secondly, although standard QUM indicators were used, some were probably measured differently across studies. Thirdly, for some countries there were only one or two QUM indicators measured. This gave a less robust picture of overall QUM and was the reason for use of a composite QUM indicator that allowed all countries to be included in the regression analyses. A few countries had outlier QUM estimates based on only 1–2 indicators; this was a possible explanation for the stronger correlation between number of policies implemented and better QUM when the analysis was confined to

**Table 4. Summary of ranking of policies and statistical conclusions from univariate analyses across three studies.**

| Policy | Policy type | Present study (Global data 2007–2011) | | | SE Asia data 2010–15 [14] | | | Global data 2003–2007 [12] | | | Overall | |
|---|---|---|---|---|---|---|---|---|---|---|---|---|
| | | Study effect estimate* | Study rank | Stat sig** | Study effect estimate* | Study rank | Stat sig** | Study effect estimate* | Study rank | Stat sig** | Sum of ranks$ | Overall rank |
| **Drugs free at the point of care** | **Economic** | **10.7** | **1** | **Yes** | **9.5** | **1** | **Yes** | **9.3** | **3** | **Yes** | **5** | **1** |
| **Government Quality Use of Medicines unit** | **Structural** | **9.8** | **4** | **Yes** | **9.0** | **4** | **Yes** | **10.9** | **1** | **Yes** | **9** | **2** |
| **Undergraduate Prescriber Standard Treatment Guideline training** | **Educational** | **5.9** | **10** | **Yes** | **9.2** | **2** | **Yes** | **10.1** | **2** | **Yes** | **14** | **3 =** |
| **Antibiotics not available Over-The-Counter** | **Regulatory** | **8.6** | **5** | **Yes** | **9.2** | **2** | **Yes** | **7.0** | **7** | **Yes** | **14** | **3 =** |
| **Generic substitution in the public sector** | **Managerial** | **10.5** | **2** | **Yes** | **4.4** | **10** | **No** | **6.6** | **9** | **Yes** | **21** | **5** |
| Drug & Therapeutic Committees in more than half of health facilities | Structural | 6.4 | 9 | Yes | 5.1 | 9 | No | 7.5 | 5 | Yes | 23 | 6 |
| National Antimicrobial Resistance Strategy | Structural | 10.2 | 3 | Yes | 1.5 | 16 | No | 7.2 | 6 | No | 25 | 7 |
| No prescriber revenue from drug sales | Economic | 7.9 | 7 | Yes | 7.8 | 6 | Yes | 3.8 | 13 | No | 26 | 8 |
| National Formulary manual updated in last 5 years | Managerial | 8.2 | 6 | Yes | 3.6 | 11 | Yes | 6.1 | 10 | Yes | 27 | 9 = |
| Public education on medicines use in last 2 years | Educational | 6.8 | 8 | Yes | 5.5 | 8 | Yes | 5.3 | 11 | Yes | 27 | 9 = |
| Generic prescribing policy in the public sector | Managerial | 2.3 | 14 | No | 8.0 | 5 | No | 4.3 | 12 | No | 31 | 11 |
| Prescription audit in last 2 years | Managerial | 5.5 | 11 | No | 7.4 | 7 | No | 3.3 | 15 | No | 33 | 12 |
| Undergraduate Prescriber Essential Medicine List training | Educational | 3.0 | 13 | No | 3.0 | 13 | No | 6.4 | 8 | Yes | 34 | 13 |
| National Drug Information Centre | Structural | 0.6 | 16 | No | -2.8 | 17 | No | 8.2 | 4 | Yes | 37 | 14 |
| No unqualified prescribers | Human resources | 3.2 | 12 | No | 2.3 | 14 | No | 3.5 | 14 | No | 40 | 15 |
| National Essential Medicine List updated in the last 2 years | Managerial | 0.9 | 15 | No | 3.2 | 12 | No | 1.9 | 16 | No | 43 | 16 |
| National Standard Treatment Guidelines updated in the last 2 years | Managerial | -3.27 | 17 | No | 1.6 | 15 | No | -0.2 | 17 | No | 49 | 17 |

\* Quantitative impact based on univariate analysis in each of the individual three studies.

\*\* 95% CI for effect estimate that did not include zero.

$ Sum of individual study ranks for each policy

countries reporting three or more QUM indicators. Finally, the clinical relevance of a composite QUM indicator is not clear, but the component indicators have relevance and we aligned each to ensure that directionality of change was constant. As with uncertainty over policy variables, any inaccuracies of medicine use estimates would likely have weakened any associations seen between policy implementation and QUM.

Our results were limited to the public sector, since there were insufficient QUM data from the private sector. While the private sector may provide most health care in many low and middle-income countries, the findings are still important since many prescribers work in both sectors and many policies are aimed at both the private and public sectors.

## Conclusions

In conclusion, repeated analyses of independent data-sets have shown replicability of two principal findings. The first is that five apparently robust essential medicines policies appear to represent the best choices for countries trying to improve medicines use, and the second one is that the implementation of multiple policies increases their effects. In 2016 The Lancet Commission on Essential Medicines identified five crucial areas of essential medicines policy. Three of these: paying for a basket of essential medicines, making essential medicines affordable and promoting quality use of medicines are strongly supported by the findings of this study [1].

## Supporting information

**S1 Table. Data on quality use of medicines by country.**
(XLSX)

**S2 Table. Data on reported policy implementation by country.**
(XLSX)

**S3 Table. Data used for study comparisons.**
(XLSX)

**S4 Table. Linear regression analyses of individual QUM indicators versus number of effective policies (out of 18) countries reported implementing.**
(DOCX)

## Acknowledgments

Richard Laing, Professor, Department of Global Health, Boston University School of Public Health, USA, formerly of WHO. He co-coordinated the medicines policy surveys in 2011.

## Author Contributions

**Conceptualization:** Kathleen Anne Holloway, Gilles Forte.

**Data curation:** Kathleen Anne Holloway, Verica Ivanovska, Solaiappan Manikandan, Mathaiyan Jayanthi, Anbarasan Mohan, Gilles Forte.

**Formal analysis:** Kathleen Anne Holloway, David Henry.

**Methodology:** Kathleen Anne Holloway, David Henry.

**Project administration:** Kathleen Anne Holloway.

**Supervision:** Kathleen Anne Holloway, Gilles Forte, David Henry.

**Validation:** Kathleen Anne Holloway, Solaiappan Manikandan, Mathaiyan Jayanthi, Anbarasan Mohan.

**Visualization:** Kathleen Anne Holloway.

**Writing – original draft:** Kathleen Anne Holloway.

**Writing – review & editing:** Kathleen Anne Holloway, Verica Ivanovska, Solaiappan Mani-kandan, Mathaiyan Jayanthi, Anbarasan Mohan, Gilles Forte, David Henry.

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
