## [Decision Letter · Decision Letter 0]

20 Nov 2019

PONE-D-19-24663

Identifying the most effective essential medicines policies: a replicability study using three WHO datasets

PLOS ONE

Dear Mr. Henry,

Thank you for submitting your manuscript to PLOS ONE. After careful consideration, we feel that it has merit but does not fully meet PLOS ONE’s publication criteria as it currently stands. Therefore, we invite you to submit a revised version of the manuscript that addresses the points raised during the review process.

We would appreciate receiving your revised manuscript by 19th December 2019. To enhance the reproducibility of your results, we recommend that if applicable you deposit your laboratory protocols in protocols.io, where a protocol can be assigned its own identifier (DOI) such that it can be cited independently in the future. For instructions see: http://journals.plos.org/plosone/s/submission-guidelines#loc-laboratory-protocols

We look forward to receiving your revised manuscript.

Kind regards,

Russell Kabir, PhD

Academic Editor

PLOS ONE

Journal Requirements:

1.

We note that you have indicated that data from this study are available upon request. PLOS only allows data to be available upon request if there are legal or ethical restrictions on sharing data publicly. For information on unacceptable data access restrictions, please see http://journals.plos.org/plosone/s/data-availability#loc-unacceptable-data-access-restrictions.

Reviewers' comments:

Reviewer's Responses to Questions

**Comments to the Author**

1. Is the manuscript technically sound, and do the data support the conclusions?

Reviewer #1: Yes

Reviewer #2: Yes

Reviewer #3: Yes

2. Has the statistical analysis been performed appropriately and rigorously? 

Reviewer #1: N/A

Reviewer #2: I Don't Know

Reviewer #3: Yes

3. Have the authors made all data underlying the findings in their manuscript fully available?

Reviewer #1: Yes

Reviewer #2: Yes

Reviewer #3: Yes

4. Is the manuscript presented in an intelligible fashion and written in standard English?

Reviewer #1: Yes

Reviewer #2: Yes

Reviewer #3: Yes

5. Review Comments to the Author

Reviewer #1: The paper entitled ``Identifying the most effective essential medicines policies: a replicability study using

three WHO datasets'' is interesting and well organized. Therefore I recommend this paper for publishing in PLOS ONE.

Reviewer #2: 1. In the abstract part, better to avoid abbreviation.

2. In abstract, you mentioned, we compared QUM scores in countries that did or did not implement specific policies and regressed QUM composite scores on the numbers of policies implemented. If the country did not implement medicine policy, what did you compare? Please either justify or clarify.

3. In abstract, the justification of conducting this study is not clear. Since, you told us there is already WHO medicine policy and countries may have drafted their own accordingly. So, what is the importance of your current study? Do you want to know the level of WHO medicine policy implementation or ? since the best medicine policy depends on the countries underlying condition.

4. strictly follow the journal guideline

Reviewer #3: Major Comments:

1. On-Page 4, Indicator Section: You mentioned that you had excluded the medicine policies from analysis if there were fewer than six countries reporting implementation or non-implementation of the policy. So, why have not you taken the policies if there were fewer than six countries reporting implementation or non-implementation of the policy? Please justify.

2. The article would be significantly improved if you were to provide a PRISMA flow diagram to map out the number of policies identified, included, and excluded and the reasons for exclusions which you have written and described (on-page 6) in your results section already.

3. On-Page 7, Strength of associations for individual policies (present study) Section: The range of estimated effects of policies you have calculated for managerial policies that are “2.3% to 10.5%” does not correspond with the Table 3 instead I found 2.8% to 10.5% as per your Table. Please recalculate the range for the estimated effects of these managerial policies.

Minor Comments:

1. On-Page no. 8: A wrong spelling has been found. Omit “standard treatment quidelines” and insert “standard treatment guideline”.

2. One-Page 7, On-Page 7, Strength of associations for individual policies (present study) Section; you have mentioned: “4.2% (range 3.2 to 5.1) for human resource policies”. Please omit this and correct this statement as “4.2% (range 3.2 to 5.1%) for human resource management policies”.

6. PLOS authors have the option to publish the peer review history of their article (what does this mean?). If published, this will include your full peer review and any attached files.

Reviewer #1: No

Reviewer #2: Yes: Seifadin Ahmed

Reviewer #3: No

---

## [Author Response · Author response to Decision Letter 0]

20 Dec 2019

Dear Editor and Reviewers

Thank you for these helpful comments. Please find our responses to each of your points below:

Editors comments

Data Availability

A. All data used for the analyses shown in this article are shown in three supplementary excel files, labelled tables S1 (QUM data), S2 (Policy data) and S3 (study comparison data) corresponding to the descriptions in the manuscript. An extra excel file showing the data used to compare policy impact on QUM in this study and the previous two published studies has now been included (table S3).

Data not shown

A. The data previously referred to as ‘not shown' are now provided. These refer to the correlation coefficients between the composite QUM scores and number of policies reported implemented for countries below and above the median Gross National Incomes. In addition, for greater transparency and clarity of the results we have moved the two supplementary figures showing the scatter-grams of QUM score vs number of policies and % upper respiratory tract infection cases treated with antibiotics vs number of policies into the main manuscript.

Reviewer Comments

Reviewer #1: 

The paper entitled ``Identifying the most effective essential medicines policies: a replicability study using three WHO datasets'' is interesting and well organized. Therefore I recommend this paper for publishing in PLOS ONE.

A. Thank you.

Reviewer #2: 

1. In the abstract part, better to avoid abbreviation.

A. We have now included the full form for all abbreviations used in the abstract

2. In abstract, you mentioned, we compared QUM scores in countries that did or did not implement specific policies and regressed QUM composite scores on the numbers of policies implemented. If the country did not implement medicine policy, what did you compare? Please either justify or clarify.

A. As the reviewer states we compared QUM scores in countries that did or did not report implementation of specific policies. In these analyses the policy was the unit of analysis and the outcome was the QUM score. So, countries that did not implement specific policies were the control group in this analysis. In the regression analyses countries were the units of analysis. The independent variable was the number of policies implemented and the outcome was a composite QUM score. Two countries reported implementing no EM policies – they were included in the analyses and are identified in the scatter-grams that are now part of the main paper.

3. In abstract, the justification of conducting this study is not clear. Since, you told us there is already WHO medicine policy and countries may have drafted their own accordingly. So, what is the importance of your current study? Do you want to know the level of WHO medicine policy implementation or? since the best medicine policy depends on the countries underlying condition.

A. The sentence justifying the study has been amended to explain that government implementation of essential medicines policies is often suboptimal and there is limited information on which policies are most effective. WHO Essential Medicines Policy consists of many different specific policies, some of which may be more or less effective in promoting quality use of medicines. The aim of this study was to identify which specific essential medicines policies were most strongly and consistently associated with better quality use of medicines. To better reflect the aim of the study, we have modified the title to read “Identifying the most effective essential medicines policies for quality use of medicines: a replicability study using three World Health Organization (WHO) data-sets”. 

4. strictly follow the journal guideline

A. We have attempted to do this.

Reviewer #3: 

Major Comments:

On-Page 4, Indicator Section: You mentioned that you had excluded the medicine policies from analysis if there were fewer than six countries reporting implementation or non-implementation of the policy. So, why have not you taken the policies if there were fewer than six countries reporting implementation or non-implementation of the policy? Please justify.

A. Our original paper in PLOS Medicine provides a more detailed account of the methods we used. The weaknesses of the data were also described and discussed in that paper. https://journals.plos.org/plosmedicine/article?id=10.1371/journal.pmed.1001724

At that time, we made a decision to deal with data sparsity and unstable statistical estimates by excluding policies from analyses where fewer than 6 countries reported implementation (or not). This was pragmatic and was considered and accepted by the editors and reviewers of that paper. Exploring reproducibility of findings was a key objective of this study so we used the same exclusion criteria in the present work. 

2. The article would be significantly improved if you were to provide a PRISMA flow diagram to map out the number of policies identified, included, and excluded and the reasons for exclusions which you have written and described (on-page 6) in your results section already.

A. The policies identified in this article all came from a questionnaire sent to Ministries of Health, not from a review of the literature. Consequently, the number of data-sets was known and our sample was complete. The supplementary file Table S2 describes the source of the results in the public domain (i.e. the WHO websites). The QUM data came from a WHO database of medicines use surveys and the search strategy has already been described elsewhere (see references 3, 10, 11) and supplementary file Table S1 describes all the articles from where the QUM data was extracted. Since this article does not describe a systematic review of the literature, but a targeted retrieval of known survey data and articles relevant to a known independent variable (policy implementation surveys) we did not feel a PRISMA flow diagram was appropriate for this article. Additionally, we were not asked for this in the original PLOS Medicine article or in the subsequent reports.

3. On-Page 7, Strength of associations for individual policies (present study) Section: The range of estimated effects of policies you have calculated for managerial policies that are “2.3% to 10.5%” does not correspond with the Table 3 instead I found 2.8% to 10.5% as per your Table. Please recalculate the range for the estimated effects of these managerial policies.

A. We apologise and have now corrected this error. The result in the manuscript was correct and table 3 has now been corrected and all the other results also checked.

Minor Comments:

1. On-Page no. 8: A wrong spelling has been found. Omit “standard treatment guidelines” and insert “standard treatment guideline”.

A. We checked the manuscript for consistency of spelling of Standard Treatment Guidelines and its acronym STGs and have standardised on this convention. 

2. One-Page 7, On-Page 7, Strength of associations for individual policies (present study) Section; you have mentioned: “4.2% (range 3.2 to 5.1) for human resource policies”. Please omit this and correct this statement as “4.2% (range 3.2 to 5.1%) for human resource management policies”.

A. Thank you. This has been corrected.

Yours Sincerely,

Kathleen Holloway and David Henry on behalf of all authors.

---

## [Editor Report · Decision Letter 1]

10 Jan 2020

Identifying the most effective essential medicines policies for quality use of medicines: a replicability study using three World Health Organisation data-sets

PONE-D-19-24663R1

Dear Dr. Henry,

We are pleased to inform you that your manuscript has been judged scientifically suitable for publication and will be formally accepted for publication once it complies with all outstanding technical requirements.

With kind regards,

Russell Kabir, PhD

Academic Editor

PLOS ONE
---

## [Editor Report · Acceptance letter]

21 Jan 2020

PONE-D-19-24663R1 

Identifying the most effective essential medicines policies for quality use of medicines: a replicability study using three World Health Organisation data-sets 

Dear Dr. Henry:

I am pleased to inform you that your manuscript has been deemed suitable for publication in PLOS ONE. Congratulations! Your manuscript is now with our production department. 

With kind regards,

on behalf of

Dr. Russell Kabir 

Academic Editor

PLOS ONE